# Muffin: Muffled Audio Encoding with Filter-Based Masking

Marcel A. Vélez Vásquez[1]

[1]Music Cognition Group, ILLC, University of Amsterdam

## Abstract

Masked autoencoders have advanced representation learning in vision and language, yet audio remains dominated by spectrogram-based approaches which then are treated like images, disregarding audio-specific characteristics. We propose *Muffled Audio Encoding*, a framework for self-supervised learning directly on raw waveforms using 1D transformers and masking through time-domain filters (e.g., low-, high-, and band-pass). This approach encourages representations that capture long-range and frequency-selective dependencies without requiring Fourier transforms and losing phase information. We outline our design and experimental plan for evaluating this method across multiple audio domains.

## 1 Introduction

In recent years, self-supervised pretraining tasks have become central to learning effective representations for language [1], images [2], and audio [3]. Among them, *masked prediction* tasks such as BERT's masked language modeling [4] and Masked Image Modeling (MIM) [5] have shown that reconstructing missing information from context can yield transferable features.

However, masked modeling has not yet been fully adapted to the unique structure of audio. Existing waveform-based methods, such as Wav2Vec 2.0 [6], typically mask contiguous segments in the *time domain*, which helps the model learn temporal dependencies but ignores rich frequency relationships that define many audio phenomena. Spectrogram-based approaches[7, 8] instead operate in the time–frequency plane but often treat spectrograms as if they were ordinary images. However, unlike images,where both spatial axes represent the same modality,spectrograms have fundamentally different axes: time and frequency. Consequently, such models may overlook the asymmetric structure of audio data and, in most cases, disregard phase information that is crucial for faithful signal reconstruction. Audio signals exhibit complex dependencies across both time and frequency. By masking in the *frequency domain*, models can learn to reconstruct missing spectral information and capture phenomena such as harmonics, timbral structure, and spectral envelopes—crucial for music, speech, and environmental sounds. To exploit this, we propose a method that performs masking directly in the waveform domain through audio **filters** (low-, high-, band-pass, and band-stop), effectively suppressing selected frequency regions while maintaining phase continuity.

We term this approach *MUFFIN* (**Muff**led au**di**o e**n**coding), which bridges time- and frequency-domain masking through filter-based perturbations within a masked autoencoding framework. We hypothesize that frequency-selective masking fosters more general and perceptually grounded representations, transferable across tasks such as speech recognition, music analysis, and environmental sound classificatio

## 2 Related work

Self-supervised learning has reshaped audio representation learning by leveraging masked or contrastive objectives without explicit labels. Most approaches adapt ideas from language or vision domains but differ in what they mask and how they represent audio.

**Masked prediction in language and vision.** BERT [4] introduced masked language modeling with low masking ratios ( 15%) for discrete tokens. Vision Masked Autoencoders (MAE) [5] extended this idea to continuous signals, masking up to 75% of image patches and reconstructing missing pixels with an asymmetric encoder–decoder design. The MAE formulation emphasizes spatial redundancy and efficient pretraining.

**Time-domain masking for speech.** **Wav2Vec 2.0** [6] learns contextualized speech embeddings by predicting quantized latent representations of masked time segments. The model is trained on raw waveforms but relies on contrastive objectives and masking entire sequences. **WavLM** [9] extends this to multi-task pretraining with denoising.

**Spectrogram-domain masking.** **Mockingjay** [7] apply masked prediction on spectrograms, typically masking 10–20% of time–frequency patches. These methods benefit from the structured representation of spectrograms but discard phase information and introduce fixed spectral resolution.

The more recent **Masked Spectrogram Modeling (MSM-MAE)** [8] adapts vision MAE to 2D spectrogram inputs, showing strong transfer to HEAR 2021 tasks.

**Bridging domains.** While waveform-based models retain temporal precision, they typically mask only contiguous time spans. Spectrogram-based models can mask across both time and frequency but often disregard phase information and treat both axes as the same modality. Our method bridges these perspectives by *masking directly in the waveform domain through filtering*: rather than masking discrete spectrogram patches, we apply parametric low-, high-, band-pass, or band-stop filters as structured masks. This removes specific frequency bands in a continuous manner while preserving temporal coherence and encouraging representations that capture both long-range and frequency-selective dependencies.

## 3  method

We employ a similar architecture to the Vision transformer, where 2D convolution-layers are replaced by 1D ones. Each input waveform segment is randomly *muffled* using filters drawn from, or a combination of:

- low-pass: remove high-frequency content;

- high-pass: remove low-frequency content;

- or, band-pass / band-stop: isolate or suppress mid-range content

The model is trained to reconstruct the original waveform from these masked variants, promoting robustness to spectral variation and encouraging multi-scale temporal representations.

## 4  Experimental Setup

**Pretraining and study design.** We begin with pretraining on the **MagnaTagATune (MTT)**[10] dataset to systematically explore the effect of **masking ratio**, **decoder depth and width**, and **filter sampling strategies**. These ablations test whether filter-based masking can yield competitive representations on modest data scales. We include basic augmentations such as random gain, polarity inversion, and time-stretching to evaluate robustness. After validating configurations on MTT, we scale pretraining to **AudioSet**[11]( 2M 10-second clips) to study how representation quality scales with dataset size and domain diversity.

**Evaluation and transfer learning.** Evaluation is conducted through **transfer learning** on the **Jukemir**[12] benchmark, which provides standardized tasks across musical attributes:

- **MTT**[10]: music tagging (multi-label classification)

- **GiantSteps**[13]: musical key estimation

- **GTZAN**[14]: Genre classification

- **EmoMusic**[15]: valence–arousal regression

We further include **OpenMIC**[16] & **NSynth**[17] for instrument recognition.

**Cross-domain evaluation.** Beyond Jukemir[12], we plan comparisons on broader audio domains:

- **Environmental sounds:** ESC-50[18] and UrbanSound8K (US8K)[19]

- **Speech:** SPCV2[20], VC1[21], and CREMA-D[22] for speaker and emotion recognition

Testing whether frequency-selective masking generalizes beyond music.

**Baselines and comparisons.** We compare against a range of pretrained self-supervised and contrastive models:

- **Waveform-based:** Wav2Vec 2.0[6], WavLM[9], ATST-Base[23], TUNe[24]

- **Spectrogram-based:** MSM-MAE[8], M2D[25], MATPAC++[26]

- **Retrieval-based:** SLAP[27] and Golden Retriever[28]

All baselines are only compared on the datasets of the original papers.

## 5  Future Work

We will next carry out the experiments outlined above, implementing MUFFIN with large-scale pretraining and executing the full set of comparisons across self-supervised, contrastive, and retrieval-based baselines. These experiments will analyze the influence of filter type, masking ratio, and model capacity on representation quality and examine whether these factors behave differently for audio than they do in vision-based masked autoencoding. We will also evaluate MUFFIN representations on the HEAR challenge to assess general-purpose audio performance, but due to the two-page limit, we omit the full list of evaluation tasks here.

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
