# OpenReview forum: "Muffin: Muffled Audio Encoding with Filter-Based Masking"
_NLDL.org/2026/Abstracts_Track — NLDL 2026 Abstracts_

### Official Review · Reviewer_PHzZ · 2025-10-30

**Soundness:** 3
**Correctness:** 2
**Rating:** 2
**Confidence:** 4

**Summary:**

Representation learning of audio data often relies on spectrograms that are treated as images. This is problematic as images have two similar dimensions while spectrograms have frequency and time dimensions. This extended abstract proposes a masked autoencoder framework that uses a specific set of maskers that work in the frequency domain. Furthermore, it proposed an extensive experimental setup.

**Strengths:**

- (Unsupervised) representation learning of different modalities (here audio) is a highly relevant topic for NLDL.
- The problem setting is well motivated.
- The proposed idea makes a lot of sense.
- The experimental setup is extensive and meaningful.

**Weaknesses:**

- The related work section (25% of the paper) is rather large. Apart from that, some things are repeated multiple times. In the context of an extended abstract, this is problematic because...
- The method section is *just* ten lines long. Model details are missing. What is the proposed/envisioned architecture? How exactly is the masking being done? This important section is way to unspecific. This is sadly a missed opportunity.
- While the proposed experimental setup is extensive and meaningful, there are no preliminary results presented which could be discussed.


Minor issues:
- Why are all baselines only compared on the datasets of their original papers (lines 166-167)? Does that make sense? Why can't they be compared on new datasets and settings?
- How big is MTT (line 126) compared to AudioSet (line 135)?
- How is the masking effect being explored/evaluated/validated (lines 127, 134)?
- reference of HEAR 2021 is missing
- many missing spaces (everywhere)
- inconsistent use of capitalization (e.g., lines 072, 092, 110, 111, 145)
- inconsistent use of tenses (Section 2)

---

### Official Review · Reviewer_ju74 · 2025-10-31

**Soundness:** 4
**Correctness:** 4
**Rating:** 5
**Confidence:** 4

**Summary:**

The paper proposes a new self-supervised learning approach for audio that designs masking strategies to be modality-aware with respect to audio’s inherent time and frequency axes. The method aims to integrate the benefits of existing time-domain and frequency-domain masking paradigms, yielding representations that are sensitive to both structures while retaining robustness. The manuscript includes a thorough literature review, detailed methodological exposition, a plan for future experiments, evaluations across multiple datasets and tasks, ablations, and comparisons with multiple models.

**Strengths:**

- Comprehensive literature review: The paper clearly surveys current approaches, articulates their limitations, and positions the proposed method within the landscape of self-supervised audio learning.
- Methodological clarity: It explains how the proposed masking integrates advantages from time- and frequency-focused strategies and builds on prior work.
- Broad evaluation and ablations: The authors report experiments across multiple datasets and tasks, and conduct ablation studies that probe the contributions of key components.
- Comparative scope: The work compares against multiple existing models.
- Forward-looking plan: The detailed outline of future experiments and objectives indicates a thoughtful roadmap for further validation and extensions.

**Weaknesses:**

- Non-unified benchmarking: Model comparisons are conducted on the datasets used in each baseline’s original paper. This practice, while convenient, undermines fairness and comparability due to differing preprocessing, splits, and evaluation protocols across studies.
- Reproducibility gaps: The paper lacks details on data splits and preprocessing. Without standardized, clearly documented splits and pipelines, results are difficult to replicate and may not be directly comparable across models.

---

### Official Review · Reviewer_A4PC · 2025-11-03

**Soundness:** 3
**Correctness:** 3
**Rating:** 4
**Confidence:** 4

**Summary:**

The abstract aims to adapt the popular masked autoencoder structure to audio encoding while avoiding the transformation of the problem into an image masking problem, which is common in this area. The author suggests using low, high, and band-pass filters for the masking tasks. So rather than masking a segment of the time series, the model aims to reconstruct the original time series, given only a filtered version of the data.

**Strengths:**

The reasoning and motivation are strong, and I believe that designing time-series-specific methods for time-series analysis is an interesting and underexplored area, which avoids the common approach of adapting the problem to existing models.

The experimental setup is well-structured and planned, including datasets and baselines.

**Weaknesses:**

The motivation for the paper is sound, and the idea behind the method is simple but could prove effective and should be explored. The work does not suggest a structural change to the masked autoencoder, but rather a slight change to the task it is presented with.

Capitalize M in method header.
Missing "n" in "classification" at end of introduction.

"All baselines are only compared on the datasets of the original papers." This sentence implies that you will not test the models yourself, but rather just copy the results. I think it would be better if you explained a bit more of your reasoning, and that stating this without argument would not be acceptable in a full paper. I also believe having a fair comparison between your method and the baselines becomes very difficult if you do not test the other methods yourself and ensure equal preprocessing, and that the data used is exactly the same.

**Suggestions and general comments that might have been left out due to the 2 page limit:**

The performance of the model suggested will be highly dependent on the filter thresholds, some mention of how these are set should be included and I imagine they could vary considerably depending on the dataset.

Patching is very common in both vision and time series models for MAE, and depending on the task, I think it can have a relatively big effect on the results, especially if the values can easily be "guessed" from the adjacent patches.

The entire task could also be reformulated by splitting the time series into three "bands", high, low and mid frequencies and then just use the three bands as separate features. Could it be interesting to look at how masking different portions more affects down stream tasks?

---

### Decision · Program_Chairs · 2025-11-05

**Decision:**

Accept

**Comment:**

The reviewers found the abstract borderline, yet the PCs believe it will be of interest to the community and should have the opportunity be presented.